**Data Availability Statement:** All relevant data are within the manuscript and its Supporting Information files.

# The yield of community-based tuberculosis and HIV among key populations in hotspot settings of Ethiopia: A cross-sectional implementation study

Z. G. Dememew[1]*, D. Jerene[1], D. G. Datiko[1], N. Hiruy[1], A. Tadesse[2], T. Moile[3], D. Bekele[4], G. Yismawu[5], K. Melkieneh[1], B. Reshu[1], P. G. Suarez[6]

**1** Management Sciences for Health (MSH), Challenge TB Project, Addis Ababa, Ethiopia, **2** Harari Health Bureau, Harar, Ethiopia, **3** Dire Dawa Health Bureau, Dire Dawa, Ethiopia, **4** Oromia Regional Health Bureau, Addis Ababa, Ethiopia, **5** Amhara Regional Health Bureau, Bahirdar, Ethiopia, **6** Management Sciences for Health, Arlington, Virginia, United States of America

* tsinatzewdu@gmail.com

## Abstract

### Objective

To determine the yield of tuberculosis (TB) and the prevalence of Human Immuno-deficiency virus (HIV) among key populations in the selected hotspot towns of Ethiopia.

### Methods

We undertook a cross-sectional implementation research during August 2017-January 2018. Trained TB focal persons and health extension workers (HEWs) identified female sex workers (FSWs), health care workers (HCWs), prison inmates, homeless, internally displaced people (IDPs), internal migratory workers (IMWs) and residents in missionary charities as key and vulnerable popuaiton. They carried out health education on the importance of TB screening and HIV testing prior to recruitment of the study participants. Symptomatic TB screening and HIV testing was done. The yield of TB was computed per 100,000 background key population.

### Results

A total of 1878 vulnerable people were screened, out of which 726 (38.7%) presumptive TB cases and 87 (4.6%) TB cases were identified. The yield of TB was 1519 (95% CI: 1218.1–1869.9). The highest proportion (19.5%) and yield of TB case (6,286 (95% CI: 3980.8–9362.3)) was among HCWs. The prevalence of HIV infection was 6%, 67 out of 1,111 tested. IMWs and FSWs represented 49.3% (33) and 28.4% (13) of the HIV infections, respectively. There was a statistically significant association of active TB cases with previous history of TB (Adjusted Odds Ratio (AOR): 11 95% CI, 4.06–29.81), HIV infection (AOR: 7.7 95% CI, 2.24–26.40), and being a HCW (AOR: 2.42 95% CI, 1.09–5.34).

**Funding:** This study was made possible by the support of the American people through the US Agency for International Development (USAID). The contents of this article are the responsibility of the authors alone and do not necessarily reflect the views of USAID or the US government.

**Competing interests:** The authors have declared that no competing interests exist.

## Conclusions

The prevalence of TB in key populations was nine times higher than 164/100,000 national estimated prevalence rate. The prevalence of HIV was five times higher than 1.15% of the national survey. The highest yield of TB was among the HCWs and the high HIV burden was detected among the FSWs and IMWs. These suggest a community and health facility based integrated and enhanced case finding approaches for TB and HIV in hotspot settings.

## Introduction

The incidence of TB has dropped in most regions of the world, including in Ethiopia [1]. However, the disease is largely concentrated in vulnerable or socially excluded populations and high-risk settings [2]. If vulnerable populations are not put at the forefront of any intervention, they will continue to be among the missed TB cases [1,2].

Ethiopia was estimated to miss 58,893 (34%) of expected TB cases in 2018 [3]. According to 2010 unpublished Ethiopian TB report, about 59% of the missed TB cases could be due to failure to detect TB in the community and among vulnerable populations. In order to identify the missed TB cases, the country has adopted the global target to identify at least 90% of TB cases among key populations by 2025 [4]. Hence, the national TB program prepared an operational guide and implementation plan on key affected populations for TB in 2017 [5].

Key and vulnerable populations for TB are defined and identified based on increased risk of TB disease due to biological and socioeconomic factors, lack of access to health services for diagnosis and treatment, and the experience of human rights violations [5]. Country-specific situations can be useful in defining key and vulnerable groups for TB [6]. Accordingly, Ethiopia has identified people living with HIV, people with diabetes, children, elders, prisoners, university residents, contacts of TB patients, miners or internal migratory workers (IMWs), cross-border refugees, internally displaced people (IDPs), homeless, female sex workers (FSWs), HCWs as key and vulnerable populations for TB [5].

On the other hand, there has been slowing or stabilizing general HIV epidemic over the last decade in Ethiopia[7]. The prevalence of HIV was 1.5% in 2011 [8] and decreased to 1.15% in 2018 [9]. The estimated number of deaths declined from 11,000 in 2015 to 5,000 in 2018 [9]. Also, the rate of TB/HIV co-infection significantly decreased from 18% in 2012 to 7% in 2017 [8,9]. Nevertheless, the burden of HIV in Ethiopia remained to be congregated in the hotspot settings such as urban areas and big cities [7,10]. Besides, the HIV epidemic is slowly rising among the high-risk population groups such as FSW and their partners [11]. Hence, it is paramount to assess HIV among the key populations in the hotspot settings of Ethiopia to deal with HIV and HIV related TB. This could contribute to the achievement of the three 90's of the global targets for both TB [4] and HIV [12].

Less evidence exists, however, about the prevalence of TB and HIV among the key and vulnerable population groups in Ethiopia. Moreover, it is essential to identify vulnerable populations based on their context [6] i.e in the hotspot setting for HIV and TB. Therefore, this study tried to determine the yield of enhanced TB case finding and the prevalence of HIV among the vulnerable population at the selected hotspot settings of Ethiopia.

## Materials and methods

### Settings

Ethiopia is the second-most populous nation in Africa, with a population of about 110 million [13]. It ranks 10th among the 30 high-TB-burden countries, with TB incidence of 164/100,000

in 2018 [3]. Harar, Dire Dawa, Woldiya, Shakiso, and Adola were the five towns in Ethiopia with a TB/HIV co-infection rate higher than 10% [14–16] (S1 Fig). These towns contribute to 2.7% of the national population. Geographic clustering of high-risk sub populations exists in these towns due to gold mining, factories, poverty, and cash crops, paving the way for high TB and HIV transmission—hence the term hotspot setting.

## Study design and interventions

We undertook a cross-sectional implementation research during August 2017-January 2018 with funding from the US Agency for International Development under the Challenge TB project. In Ethiopia, Challenge TB provided support to the national TB program in nine of the 11 administrative regions. We initiated the study after consultation with regional, zonal, and district TB focal persons. The project built the capacity of the program managers, HCWs, and HEWs on TB and TB/HIV screening, diagnosis and treatment. The project also technically and financially assisted supportive supervisions and program reviews on TB and TB/HIV. Improving sputum specimen and patient referral system, and strengthening data quality and reporting system through the district health information system (DHIS) were another support issued to the national TB program by Challenge TB project.

## Identification of hotspot settings and key populations

We selected the five study towns as the hotspot for TB and HIV because of their higher TB/HIV co-infection rate [14–16] as compared to other towns in the country. All the missionary residents, hotels, mining or construction offices, correctional facilities, health facilities, street tukuls of the homeless and refugee centers in the five towns were selected as sites of the data collection.

TB focal persons, HCWs that coordinate comprehensive TB and TB/HIV activities, and HEWs—the female community workers employed to execute the health extension program— were trained on the procedures of defining [5,6 &13], identifying and sampling the key populations in the data collection areas. They were also trained on the information they need to deliver during health education to the key population before recruitment; such as TB transmission, purpose of the study and the advantages of being screened for TB and HIV. Hence, they carried out 15–20 minutes of health education before recruitment and data collection among the key population to enhance the TB screening and HIV testing.

They recruited FSWs at the hotels after obtaining permission from the owner of the hotel. The HEWs and TB focal persons deployed homeless individuals in the street. IDP, HCWs, prison inmates and IMW were recruited after the heads of the offices of road construction and mining (for IMW), health care facility (for HCWs), correctional facility (prisoners) and refugee centers (IDPs) were approached. The HCWs were clinicians, such as registered nurses, interns, medical doctors and public health officers that were involved in managing patients in public health facilities found in the study towns. IDPs were those individuals that were displaced from Somalia region of the country to Harari region due to ethnic conflict during the study period. The IDPs arrived at refugee center near Harar town 4–6 weeks prior to data collection.

The key population that understood the objective of the study and willing to participate after the health education, and in a relative good health status were involved in the study. Those who understood the aim of the study but refused to be part of the study and/or were sick during the data collection were excluded (Fig 1).

## Sampling the key population

There were a total of 3,400 prison inmates, 250 residents of facilities operated by charities and 350 HCWs reported from the five towns' health office. These numbers were taken as a

sampling frame. One-third of the sampling frame from each of these key population was randomly selected to be a study population in the five towns. Excel sheet was used to undertake the simple random selection of the study population from the sampling frame for the prison inmates, missionary residents and HCWs.

However, the number for FSWs, IMWs, IDPs and homeless could not be obtained. Hence, the sampling frame was established during the data collection. That is, when the HEWs and TB focal persons visited the FSWs in the hotels, IMWs at the workplace, IDPs in the refugee centers and homeless in the street, they registered these key populaiton on excel sheet (sampling frame). Then one-third of the sampling frame was randomly selected to be the study participants. Accordingly, there were 639 FSWs, 730 IMWs, 315 IDPs, and 55 homeless individuals that were registered as sampling frame from where random sampling was carried out during the study period.

All in all, 5729 key populations were taken as sampling frame or background key population in the five study towns. A total of 1929 vulnerable population were selected randomly and approached for TB screening; 1125 prison inmates, 87 residents of facilities operated by charities, 123 HCWs, 225 FSWs, 245 IMWs, 105 IDPs and 19 homeless individuals. About 1878 (97%) of them accepted the screened for TB; 221 FSWs, 237 IMWs, 1112 prison inmates, 79 residents of facilities operated by charities, 113 HCWs, 102 IDPs, and 14 homeless (Fig 1).

## Screening and diagnosis of TB, and HIV testing among the key population

The HEWs and TB focal persons carried out symptom-based screening for TB. At the same time, they did confidential HIV testing and counseling. An individual having cough, fever, and night sweating of more than two weeks or weight loss were taken as a presumptive TB case or positive screening test [17]. Nationally approved rapid HIV test kits were used for HIV testing. The identified presumed TB cases and HIV positive key populations were referred to the health facilities in study towns having TB DOTS and chronic HIV care services. These were the facilities with external quality control for diagnostic tests. At the health facilities, depending on their complaint, the presumed TB cases underwent clinical evaluation (history and physical examination), acid-fast bacilli (AFB) test, Gene X-pert test, fine needle aspiration (FNA) or chest X-ray. For a single key population, the time spent for recruitment, TB screening, HIV testing and counseling, and recording all the outcomes of TB screening and evaluation ranged from 15–20 minutes.

TB cases were categorized as bacteriologically confirmed pulmonary TB (PTB) cases where the diagnosis is made using AFB or Gene X-pert, clinically diagnosed smear-negative PTB if the diagnosed is based on clinical findings, and clinically diagnosed extra-PTB (EPTB) if the diagnosis is based on clinical evidence and the disease is out of the lung. The classification of TB was also made as drug-susceptible TB if the disease is responding to first-line anti-TB drug and Multi-drug resistance or rifampicin resistance TB (MDR-TB/RR-TB) if the TB disease is resistance at least to rifampicin and isoniazid [17]. The key populations that had already known their HIV status were also recorded.

### Data source

The study coordinators prepared a register of key and vulnerable populations which the TB focal persons and HEWs used to record the number of people approached for TB screening, their sociodemographic characteristics, presumed TB cases, TB cases, and HIV status. The register of key and vulnerable population captured the information obtained from the individual key population and the outcomes of the clinical evaluation and laboratory investigations.

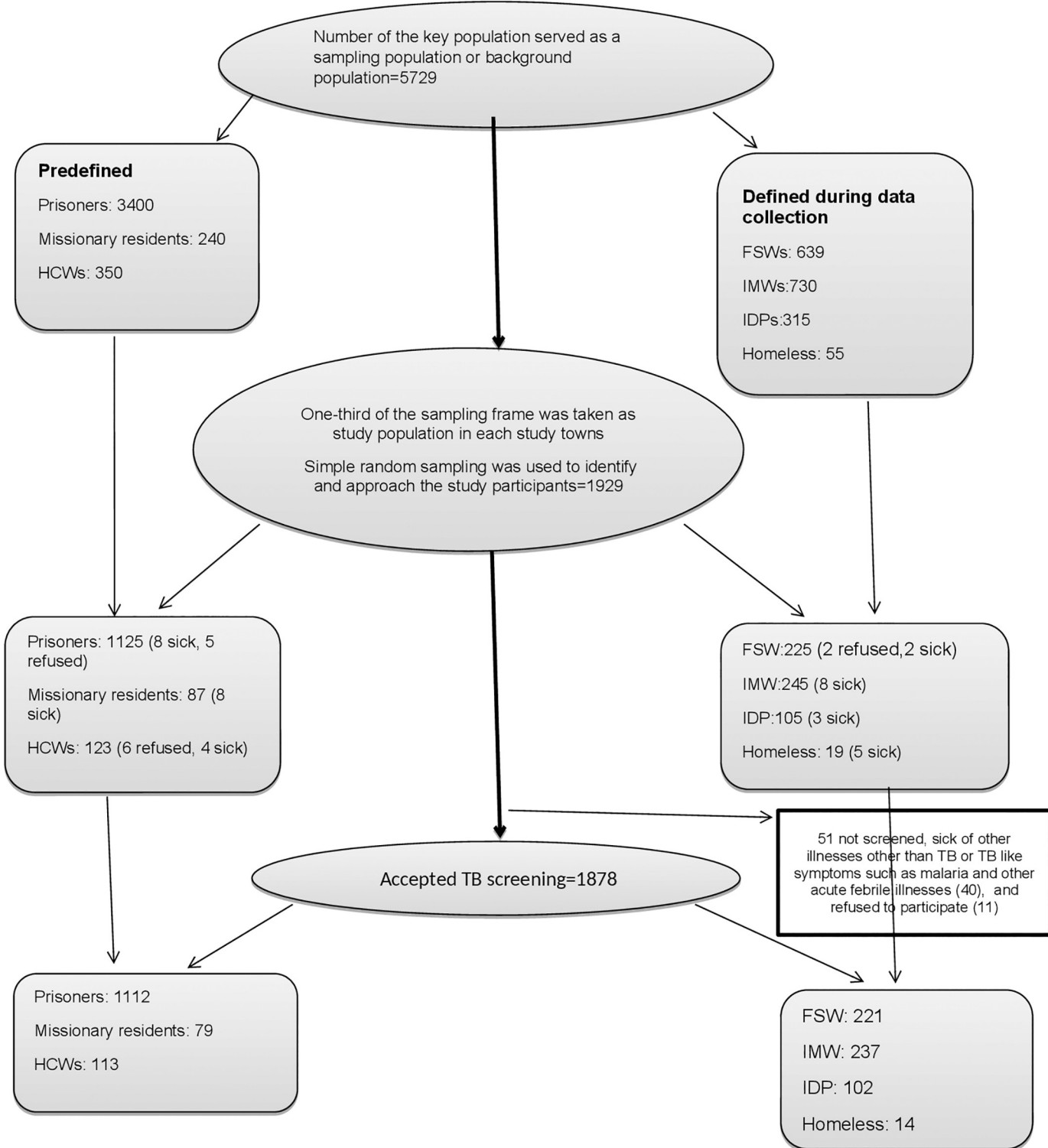

**Fig 1. The flow chart shows sampling and selection of the key population.** Predefined key population are those whose number had already been documented and reported by the study towns' health offices. The **"defined during data collection"** key population were those whose number were not known by the study towns' health office and their background population was determined during the data collection period. The sampling frame was taken as the total key population in the study towns, and thus the background population to compute the yield or prevalence of TB. Bold arrow is to show the crude procedure one after the other; listing the sampling frame, random selection of one-third of the sampling frame, approaching for TB screening. Light arrow is to show the same procedure in each key population; FSW (female sex workers), IMW (internal migratory workers), IDP (internally displaced people), HCW (health care workers), missionary

residents are the one supported by the charity organization. Most of the refusals were from HCW and FSW for they were busy, and from prison inmates. The malaria illness and other acute febrile illness made the other key population difficult to participate in the study.

## Data quality

We deployed two trained data entry clerks. The first carried out primary data entry and the second checked for discrepancies in the data. The study coordinators also supervised data collection and data entry for consistency and completeness.

## Data analysis

The Epi Info statistical package (Version 7.2.2.16; Atlanta, Georgia: Centers for Disease Control and Prevention; 2018) was used for data entry and cleaning. We imported the data to Stata (College Station, Texas: StataCorp; 2013) for data analysis. Frequency, percentage, mean, and other descriptive statistics were used. The notified TB cases per 100,000 background key population and proportions were used to compute the prevalence of TB in the key population. Adjusting for sociodemographic characteristics in the key population, bivariate and multivariable (forward conditional) logistic regression—applying the odds ratio and 95% confidence interval (95% CI)—were used to determine factors associated with active TB in the key population. The independent variable was the presence of TB case. The dependent variables were sex (male and female), age (categorized based on the median, below and above 28 years), educational status (below high school, high school and above), marriage (married/with partner and non-married or without partner), HIV status (HIV positive and HIV negative), previous history of TB (having at least one episode of TB before and never had TB case), and type of key population (FSW, IWM, IDPs, HCWs).

## Ethical considerations

The ethical review committee (ERC) of the respective regions of the study towns approved the study protocol. These were ERC of Oromia regional health bureau, Institutional Review Board (IRB) of the Amhara public health institute, the ERC of the research wing of Dire Dawa health bureau and ERC of Harari regional health bureau. We obtained support letters from the ERC and IRB of these regions and towns to communicate with the relevant local organizations and town health offices where the key population were found. We also sought and received informed written consent from the study participants before data collection. Permission was requested from the guardian and parents in case of children. Even though a separate consent requested for TB screening and HIV screening, it was asked one after the other; first for TB screening and then for HIV testing. All the key and vulnerable population were informed that it is their full right to exit from the study if they are not willing. However, all were getting a TB screening, evaluation and treatment services irrespective of their willingness or refusal to participate in the study. That is, the respective TB case and HIV positive key population were linked to and managed at the TB DOTS and chronic HIV care of the health facilities in the study towns.

## Results

### Sociodemographic characteristics

Of the 1,878 participants approached and screened for TB, 1326 (70.6%) were men. The mean and the median age were 30.5 years and 28 years (Range: 5–80 years), respectively. About half

**Table 1. Sociodemographic characteristics of the vulnerable population in the selected five towns of Ethiopia, August 2017- January 2018.**

| Variables | Frequency | Percent |
|---|---:|---:|
| **Types of vulnerable populations** | | |
| FSWs | 221 | 11.77 |
| IMWs | 237 | 12.62 |
| Prisoners | 1,112 | 59.21 |
| Residents of missionary charity facilities | 79 | 4.21 |
| Homeless people | 14 | 0.75 |
| IDPs | 102 | 5.43 |
| HCWs | 113 | 6.02 |
| Total | 1,878 | 100 |
| **Sex** | | |
| Female | 552 | 29.39 |
| Male | 1,326 | 70.61 |
| Total | 1,878 | 100 |
| **Age in years** | | |
| < 15 | 19 | 1.0 |
| 15–24 | 594 | 31.9 |
| 25–34 | 721 | 38.7 |
| 35–44 | 334 | 17.9 |
| > 44 | 195 | 10.5 |
| Total with age determined | 1,863 | 100 |
| **Marital status** | | |
| Married | 911 | 50.36 |
| Divorced/separated | 245 | 13.54 |
| Single/never married | 636 | 35.16 |
| Widowed | 17 | 0.94 |
| Total with marital status determined | 1,809 | 100 |
| **Educational status** | | |
| Primary school (1st-6th grade) | 855 | 47.77 |
| 7th-8th grade | 628 | 35.08 |
| 9th-12th grade | 184 | 10.28 |
| 12th grade or above | 123 | 6.87 |
| Total with educational status determined | 1,790 | 100 |

of the study participants were married and had attended at least primary school (Table 1). The detail of sociodemographic characteristic of each key population is described in S1 Table.

## Screening and evaluation

One hundred and one (5.4%) of the screened vulnerable population had a history of previous TB treatment, and five of them (0.3%) were on treatment during data collection. Of the 1,878 screened, 726 (38.7%) were presumptive TB cases, of whom 210 (28.9%) were clinically evaluated, 126 (17.4%) were investigated using acid-fast bacilli (AFB) testing and 612 (84.3%) were tested by GeneX-pert. A total of 959 key population underwent at least a clinical evaluation or TB laboratory test. A total of 87 (4.6%) TB cases were identified and 65 (74.7%) were bacteriologically confirmed; 62 (95.4%) were drug susceptible TB and 3 (4.6%) were MDR-TB cases (Table 2). Note that five of the vulnerable (IMW) were already on treatment and had clinical EPTB (1) and PTB (4).

**Table 2. Tuberculosis screening, evaluation, and final status of the vulnerable population in the five towns of Ethiopia, August 2017–January 2018.**

| Variables | Frequency | Percent |
|---|---|---|
| **Previous TB episode** | | |
| Had TB once | 92 | 4.9 |
| Had TB twice | 9 | 0.5 |
| On treatment now | 5 | 0.3 |
| Never | 1,772 | 94.4 |
| Total | 1,878 | 100.0 |
| **Outcome of TB screening** | | |
| Positive | 726 | 38.7 |
| Negative | 1,152* | 61.3 |
| Total | 1,878 | 100.0 |
| **Means of TB investigation** | | |
| AFB | 126 | 13.1 |
| GeneXpert | 612 | 63.8 |
| Clinical only | 210 | 21.9 |
| Chest x-ray | 9 | 0.9 |
| FNA | 2 | 0.2 |
| Total evaluation done using at least one of the above criteria | 959 | 100.0 |
| **Outcome of the investigation** | | |
| No TB | 1,748 | 93.3 |
| TB diagnosed during the study period | 87 | 4.6 |
| Result could not be found | 38 | 2.1 |
| Total | 1,873 | 100.0 |
| **Type and site of TB** | | |
| **Drug-susceptible TB** | | |
| *Bacteriologically confirmed PTB* | 62 | 71.3 |
| *Clinical extrapulmonary TB (EPTB)* | 11 | 10.3 |
| *Clinical pulmonary TB (PTB)* | 16 | 14.9 |
| **MDR-TB (all new and bacteriologically confirned PTB)** | 3 | 3.4 |
| Total | 92 | 100.0 |

## The prevalence of TB, HIV, and TB/HIV co-infection among the vulnerable population

Of the 87 TB cases, 27 (31%) were prisoner inmates and 22 (25.3%) were HCWs. The highest proportion of TB cases was found among HCWs (19.5%) and the lowest was among IDPs (1%). Overall, the yield of TB cases per the 100k background vulnerable population was 1,519 (95% CI:1218.1–1869.9), nine times the estimated prevalence rate of 164/100k in the general population during the study period. The prevalence of TB among HCWs was the highest of all (6,286 (95% CI:3980.8–9362.3)), the least being among the IDPs (317.5 (95% CI: 80.4–1756.0)). No TB case was detected among the homeless individuals (Table 3).

About 1293 (69%) of the identified key population were approached for HIV testing and counseling; 1111 (59.2%) were tested and 183 (14.2%) refused testing. The overall prevalence of HIV infection, new plus already on treatment, was 67 out of the tested 1111 (6%), five times the 1.15% prevalence estimate in the general population. IMWs and FSWs represented 49.3% (33) and 28.4% (13) of the HIV infections, respectively (Table 4). Note that HIV positives reported here include those who had already known their status and were on HIV care.

**Table 3. Tuberculosis case finding among the vulnerable populations in the five towns of Ethiopia, August 2017–January 2018.**

| Type of Vulnerable Population | Result of TB screening and evaluation | | Total key population selected and screened | Number of background key population/ sampling frame | Proportion of TB among the screened and evaluated | Notified TB cases per 100,000 background vulnerable population (95% CI) | Comparison with the estimated TB prevalence for the general population (164/100,000) |
|---|---|---|---|---|---|---|---|
| | No TB | TB | | | | | |
| FSWs | 203 | 18 | 221 | 639 | 8.1 | 2,817 (1677.8–4415.5) | 17.2 |
| IMWs | 214 | 18 | 237 | 730 | 7.6 | 2,466 (1467.8–3869.0) | 15.0 |
| Prison inmates | 1,061 | 27 | 1,112 | 3,400 | 2.4 | 794 (524.0–1153.3) | 4.8 |
| Residents of missionary charity | 64 | 1 | 79 | 240 | 1.3 | 417 (105.5–2299.5) | 2.5 |
| IDPs | 101 | 1 | 102 | 315 | 1 | 317.5 (80.4–1756.0) | 1.9 |
| HCWs | 91 | 22 | 113 | 350 | 19.5 | 6,286 (3980.8–9362.3) | 38.3 |
| Homeless | 14 | 0 | 14 | 55 | NA | NA | NA |
| Total | 1,748 | 87 | 1,878 | 5,729 | 4.6 | 1,519 (1218.1–1869.9) | 9.3 |

Forty-nine (4.4%) of the key population with the documented HIV status had already aware that they were HIV infected and were on HIV care. Eighteen (26.9%) were new HIV-positive patients detected during the study. There were 21 (31.3%) TB cases among the HIV infected key population. HIV testing was carried out in 41 (47%) of the newly identified TB cases where 21 (51.2%) were HIV positive (S2 Table).

## Factors associated with TB cases among the vulnerable population

In the bivariate analysis, having not married, being health care work, attending the lower educational level with the previous history of TB and infection with HIV have a statistically significant association with the diagnosis of TB case among the vulnerable population. In the multivariable analysis, previous history of TB disease (AOR: 11 95%; CI, 4.06–29.81), HIV infection (AOR: 7.7 95%; CI, 2.24–26.40) and being HCW (AOR: 2.42; 95% CI, 1.09–5.34) remained statistically related to active TB (Table 5).

## Discussion

This study defined and identified FSWs, IMWs, missionary facility residents, prison inmates, IDPs, HCWs, and the homeless as key and vulnerable populations in the selected hotspot

**Table 4. The status of HIV screening, testing, and results among the vulnerable population in five towns of Ethiopia, August 2017–January 2018.**

| Type of Vulnerable Population | HIV test result [a] | | | Total key population | Total tested (%) | HIV positive (%) | Comparison with the estimated prevalence in the general population (1.15%) |
|---|---|---|---|---|---|---|---|
| | Positive | Negative | Refused | | | | |
| FSWs | 19 | 102 | 22 | 221 | 121(54.8) | 15.7 | 13.7x |
| IMW s | 33 | 127 | 51 | 237 | 160 (67.5) | 20.6 | 17.9x |
| Prison inmates | 13 | 671 | 84 | 1112 | 684 (61.5) | 1.9 | 1.7x |
| Missionary facility residents | 2 | 34 | 25 | 79 | 36 (45.6) | 5.6 | 4.8x |
| Homeless | 0 | 8 | 0 | 14 | 8 (57.1) | 0.0 | 0x |
| HCWs | 0 | 102 | 0 | 113 | 102 (90.3) | 0.0 | 0x |
| IDP | NA (note applicable) | NA | NA | 102 | NA | NA | NA |
| Total | 67 | 1,044 | 182 | 1878 | 1,111 (59.2) | 6.0 | 5.2x |

**Table 5. Bivariate and multivariable analysis of factors associated with active TB among key and vulnerable populations in the five towns of Ethiopia, August 2017–January 2018.**

| Variables | Category (% TB case) | Bivariate analysis | | Multivariable analysis | |
|---|---|---|---|---|---|
| | | Crude odds ratio (COR) | 95% CI | AOR | 95% CI |
| Sex (N = 87) | 1. Male (70.1) | 0.97 | 0.61–1.56 | 0.65 | 0.37–1.16 |
| | 2. Female (29.9) | 1 | | | |
| Age in years (N = 86) | 1. >28 years (52.3) | 1 | | | |
| | 2. >= 28 years (47.7) | 0.83 | 0.54–1.28 | 0.8 | 0.08–7.78 |
| Marital status (N = 80) | 1. Not married (73.8) | 2.98 | 1.80–4.95 | 2.09 | 0.86–5.06 |
| | 2. Married (26.2) | 1 | | | |
| History of TB (N = 87) | 1. Yes (34.5) | 11.54 | 6.93–1921 | 11 | 4.06–29.81 |
| | 2. No (65.5) | 1 | | | |
| Educational status(N = 80) | 1. Below high school (78.8) | 2.16 | 1.32–3.54 | 0.41 | 0.11–1.47 |
| | 2. High school and above (21.3) | 1 | | | |
| HIV status (N = 41) | 1. HIV+ (51.2) | 23.37 | 11.84–46.13 | 7.7 | 2.24–26.40 |
| | 2. HIV- (48.9) | 1 | | | |
| Type of vulnerable population (N = 85) | 1. HCWs (25.9) | 2.73 | 1.39–5.33 | 2.42 | 1.09–5.34 |
| | 2. IMWs (21.2) | 0.93 | 0.46–1.83 | 1.18 | 0.51–2.72 |
| | 3. Prisoners (31.8) | 0.28 | 0.15–0.52 | 0.51 | 0.24–1.06 |
| | 4. FSWs (21.2) | 1 | | | |

settings. Through the enhanced community-based TB/HIV intervention in the selected hot-spot settings, we reported an overall nine times more TB cases and five times more HIV infections among the key and vulnerable population as compared to the general population. Also, the study indicated that being HCW, having HIV infection and previous episode of TB disease seem statistically associated with the development of active TB. Hence, ending the TB epidemic in Ethiopia cannot be successful without collaboration to find and treat TB and HIV at the community level among the disproportionately affected vulnerable and key populations in hotspot settings. The implementation of comprehensive, tailored and enhanced TB case finding should be prioritized in these congregate settings.

Other studies support the finding that HCWs have a high incidence and prevalence of TB [18–20]. The yield of TB among the HCWs in this study is even higher than the studies in China [18,19]. This might be due to poor TB infection control at health facilities in Ethiopia [20]. Similarly, a systematic review in low- and middle-income countries indicated that the incidence rate of TB among HCWs is 2.94 times higher (95% CI, 1.67–5.19) than in the general population [21]. Although HCWs have the highest rate of TB, there was no detected HIV infection among them. The stigma associated with HIV disclosure [22] might make it difficult to find HIV-infected HCWs [23]. This could also indicate that it is worth looking for other determinants of the high TB incidence among HCWs though HIV remains a key risk factor for TB among HCWs in high-HIV-burden settings [24]. For instance, poor TB infection control in health facilities [25] and repeated exposure to TB infection and thus nosocomial TB infection [21] could explain the high TB prevalence rate among HCWs. This could partly due to greater attention to the TB symptoms and higher awareness of the disease among the HCWs. TB infections at health facilities could be due to unidentified and unsuspected TB cases [26] and the prolonged period prior to diagnosing TB cases [27]. Therefore, reducing delay in the diagnosis of TB in health facilities could be one of the approaches to lower the high TB burden among HCWs [27]. Practising comprehensive TB infection control could also reduce the high TB transmission in health facilities [25]. In addition, periodic clinical

evaluation and tuberculin skin tests, and provision of preventive therapy could be considered to avert the occurrence of active TB among HCWs in Ethiopia.

As compared to other vulnerable and key populations, FSWs and IMWs were found to have a high TB case notification rate in the background of higher HIV prevalence. Both are sexually active, young and usually migrate to cash crop areas. IMWs are often the clients of FSWs at mega projects and mining areas [28]. Due to their low socioeconomic and educational status, IMWs and FSWs live in congregate and overcrowded homes and practice risky sexual behavior [28]. Hence, they are at risk of contracting and transmitting both HIV and TB infections. In addition, IMWs and FSWs play a key role in the epidemics of TB and HIV. In high-burden settings, the two diseases reinforce each other and share common risk factors [29]. So, a single service provided to people with multiple related risks represents a missed opportunity to diagnose, treat, and prevent TB or HIV. The shortcomings of this approach are evident in communities that are considered vulnerable populations [30]. Establishing and strengthening the integration of TB and HIV interventions at primary health care facilities and at the community level in hotspot settings among these vulnerable and key populations is therefore critical.

As we would expect, the yield of TB among prison inmates was five times higher than in the general population. The prevalence rate of TB among inmates in this study is higher than the rate in a recent systematic review in Ethiopia [31]. This could be because the prisoners in our study came from a hotspot area where HIV could have also contributed. However, the yield of TB case is lower than the prevalence rate reported in Côte D'Ivoire where it is 10–44 times higher than the rate in the general population [32]. This difference may be due to the variation in the diagnostic facilities and TB epidemiology between the two settings. In our study, the prison inmates represented the highest number of vulnerable and key populations screened, and they contributed to the highest proportion of overall TB case notification. So, they remain the vulnerable population most in need of tailored interventions to address TB transmission in correctional and detention centers. Like IMWs and FSWs, prisoners may also serve as a reservoir of TB and could shift the TB epidemic from correctional facilities to the community [33]. Therefore, entry and exit screenings and scheduled mass screenings are worth carrying out in correctional and detention centers in Ethiopia. Nevertheless, the prevalence of HIV infection among the prisoners was slightly higher than in the general population (1.7 vs 1.2%). Yet 11% of them refused HIV testing. The risk of TB among prisoners might not be explained only by HIV infection [34,35] but could also be due to weak implementation of TB infection control [20].

The homeless were identified as one of the vulnerable and key populations, although they were few, with no TB and HIV cases identified. Evidence shows that they have a higher burden of TB [36] due to malnutrition and addiction of various kinds, such as smoking [37]. Women who live on the street are also at high risk of sexual assault and concurrent risk for HIV infection [38] and thus for TB. The study also identified a few individuals living in missionary facilities, with low TB case identification. Some of these were children, in whom TB diagnosis is usually difficult [39]. Nevertheless, the homeless and residents of charities are at risk for TB and HIV infections. Future studies involving higher numbers of these populations could complement the investigation of the prevalence of TB and HIV infection in Ethiopia.

During the study period, Ethiopia experienced unrest that displaced several people, specifically in the eastern part of the country, where two of the study towns are located. The outreach activity to IDPs detected fewer TB cases. Evidences have reported that IDPs and other refugees are at risk for TB disease [40,41]. The longer IDPs stay in refugee centers, the higher the likelihood that they will develop TB [41]. Hence, follow-on TB and HIV screening and evaluation might detect additional TB and HIV cases. This is because overcrowding, stressful living

conditions, malnutrition, and lack of access to health services could put these vulnerable people at high risk for TB [42].

The findings in this study should be interpreted cautiously, for there were limitations. Although the study included a high number of key and vulnerable populations at the community level, it should have included more homeless, refugee, cross-border refugees, children and diabetic mellitus in the other areas of the country other than the five towns. The data is skewed towards male for there were a lot of male HCWs, prison-in-mates and IDP, possibly challenging the generalization to similar settings. Only a few independent variables were considered for risk factor analysis. Thus, future studies need to investigate other risk factors for TB infection and disease in key and vulnerable populations. Besides, the shortage of the HIV testing kits made HIV testing difficult for some key population. The living situation of the homeless and the working environment of the FSW challenged the sampling and data gathering. Eventually, the crude comparison of the prevalence or notification of TB case between the national figure and the key population should consider the mixed TB case finding—of active and passive—in the general population and the enhanced active TB case finding in this study.

## Conclusions

The enhanced community-based TB/HIV activity detected more HIV and TB cases among vulnerable and key populations in hotspot settings as compared to the general population. The yield of TB among HCWs and HIV prevalence among FSWs and IMWs were significant. Therefore, mapping hotspot settings and prioritizing key and vulnerable populations at high risk for TB and HIV are essential to slow the transmission of both diseases. This suggests that ending the TB epidemic and also of HIV in Ethiopia will not be successful without community-based collaboration of TB and HIV programs among the disproportionately affected vulnerable and key populations in hotspot settings.

## Supporting information

**S1 Fig. The map of Ethiopia with the major cities and study towns: The irregular line on the map of Ethiopia shows the administrative boundary between regions.** The cities in red colors are bigger cities in Ethiopia and are also the capital cities of the regions; they had less than 10% TB/HIV co-infection. The area with green-yellow color spot are the study towns assigned as hotspot settings for the TB and HIV, with TB/HIV co-infection of at least 10%.
(DOCX)

**S1 Table. Table showing the sociodemographic details of each key population.** Age is categorized based on the median of the overall population. In the marital status, divorced and widowed were merged to show a key population without a partner. The educational category was based on grading of the Ethiopian education system. Numbers in the bracket are to show the percentage.
(DOCX)

**S2 Table. This table demonstrates HIV, TB/HIV, and linkage to ART services.** HIV counseling and request for test was offered for 1293 key populations. Testing was offered for 1111 of them. 182 refused testing, and the other 585 HIV counselling and testing was not undertaken due to the shortage of HIV testing kits. Therefore, the respective TB screened 182 and 582 key populations were not tested for HIV due to refusal and shortage of HIV screening test kits.
(DOCX)

## Acknowledgments

The authors wish to thank the Amhara, Dire Dawa, Harari, and Oromia Regional Health Bureaus for their strong collaboration with Challenge TB; and Barbara K. Timmons for editing the manuscript. This study was made possible by the support of the American people through the US Agency for International Development (USAID). The contents of this article are the responsibility of the authors alone and do not necessarily reflect the views of USAID or the US government.

## Author Contributions

**Conceptualization:** Z. G. Dememew, D. Jerene, K. Melkieneh, B. Reshu.

**Data curation:** Z. G. Dememew, N. Hiruy, D. Bekele.

**Formal analysis:** Z. G. Dememew, P. G. Suarez.

**Funding acquisition:** D. Jerene.

**Investigation:** Z. G. Dememew, A. Tadesse, T. Moile, D. Bekele, G. Yismawu, K. Melkieneh, B. Reshu, P. G. Suarez.

**Methodology:** Z. G. Dememew, D. G. Datiko, N. Hiruy, T. Moile, G. Yismawu, K. Melkieneh, B. Reshu.

**Project administration:** Z. G. Dememew, D. Jerene, D. G. Datiko, A. Tadesse, T. Moile, D. Bekele, G. Yismawu, K. Melkieneh, P. G. Suarez.

**Resources:** D. Jerene, P. G. Suarez.

**Software:** Z. G. Dememew.

**Supervision:** Z. G. Dememew, D. G. Datiko, N. Hiruy, A. Tadesse, D. Bekele, K. Melkieneh, B. Reshu.

**Validation:** A. Tadesse.

**Visualization:** T. Moile.

**Writing – original draft:** Z. G. Dememew.

**Writing – review & editing:** D. Jerene, D. G. Datiko, N. Hiruy, D. Bekele, G. Yismawu, K. Melkieneh.

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
