## [Decision Letter · Decision Letter 0]

6 Mar 2020

PONE-D-20-01766

The yield of community-based enhanced tuberculosis and Human Immuno-deficiency virus case finding among key populations in selected hotspot settings of Ethiopia: a cross-sectional implementation study

PLOS ONE

Dear Dr. Dememew,

Thank you for submitting your manuscript to PLOS ONE. After careful consideration, we feel that it has merit but does not fully meet PLOS ONE’s publication criteria as it currently stands. Therefore, we invite you to submit a revised version of the manuscript that addresses the points raised during the review process.

We would appreciate receiving your revised manuscript by Apr 20 2020 11:59PM. To enhance the reproducibility of your results, we recommend that if applicable you deposit your laboratory protocols in protocols.io, where a protocol can be assigned its own identifier (DOI) such that it can be cited independently in the future. For instructions see: http://journals.plos.org/plosone/s/submission-guidelines#loc-laboratory-protocols

We look forward to receiving your revised manuscript.

Kind regards,

HASNAIN SEYED EHTESHAM

Academic Editor

PLOS ONE

Additional Editor Comments (if provided):

Major Revision

Journal Requirements:

2. Please provide additional information regarding the considerations  made for the prisoners included in this study. For instance, please discuss whether participants were able to opt out of the study and whether individuals who did not participate receive the same treatment offered to participants.

3. Please address the following:

- Please ensure you have thoroughly discussed any potential limitations of this study within the Discussion section, for example,  the skewed nature of the dataset towards males.

- Please refer to any sample size calculations performed prior to participant recruitment. If these were not performed please justify the reasons. Please refer to our statistical reporting guidelines for assistance (https://journals.plos.org/plosone/s/submission-guidelines.#loc-statistical-reporting).

- Please modify the title to ensure that it is meeting PLOS’ guidelines (https://journals.plos.org/plosone/s/submission-guidelines#loc-title). In particular, the title should be "specific, descriptive, concise, and comprehensible to readers outside the field". In this case, we feel that the title is not concise, for example HIV could be used instead of "Human Immuno-deficiency virus".

4. Thank you for your ethics statement : "The ethical review committee (ERC) of the respective regions of the study towns approved the study protocol."

5. Please amend the manuscript submission data (via Edit Submission) to include author D. Gemechu, D. Bekele, G. Yismawu, K.Melkieneh, B. Reshu, P.G. Suarez.

6. Please amend your authorship list in your manuscript file to include author Daniel G Datiko.

7. Please include your tables as part of your main manuscript and remove the individual files. Please note that supplementary tables (should remain/ be uploaded) as separate "supporting information" files

8. We note that Figure S1 in your submission contain map images which may be copyrighted. All PLOS content is published under the Creative Commons Attribution License (CC BY 4.0), which means that the manuscript, images, and Supporting Information files will be freely available online, and any third party is permitted to access, download, copy, distribute, and use these materials in any way, even commercially, with proper attribution. For these reasons, we cannot publish previously copyrighted maps or satellite images created using proprietary data, such as Google software (Google Maps, Street View, and Earth). For more information, see our copyright guidelines: http://journals.plos.org/plosone/s/licenses-and-copyright.

1.    You may seek permission from the original copyright holder of Figure S1 to publish the content specifically under the CC BY 4.0 license. 

Reviewers' comments:

Reviewer's Responses to Questions

**Comments to the Author**

1. Is the manuscript technically sound, and do the data support the conclusions?

Reviewer #1: Yes

Reviewer #2: Yes

2. Has the statistical analysis been performed appropriately and rigorously? 

Reviewer #1: Yes

Reviewer #2: Yes

3. Have the authors made all data underlying the findings in their manuscript fully available?

Reviewer #1: Yes

Reviewer #2: Yes

4. Is the manuscript presented in an intelligible fashion and written in standard English?

Reviewer #1: No

Reviewer #2: Yes

5. Review Comments to the Author

Reviewer #1: The manuscript is a commendable attempt that reflects the approach of precision public health in the field of TB and HIV a socioeconomically weak country having an additional burden of chronic conflict. The findings should prove valuable for control efforts in similiar settings globally. There are however some shortcomings and information gaps which should be attended to.

1. The manuscript requires language editing all through the manuscript including technical termseg. line 160 (illness is out of lung tissue: meaning extrapulmonary TB) Authors are requested to take professional help in undertaking this

2. Line 97: Please describe who are the TB focal persons

3. Line 120 : It would be helpful to know what the morbidity was at the time of data collection and which study population they represented. This could be represented in the Flow Chart which incidentally needs to be enumerated as Fig.

4. Whilst acceptance of TB and HIV screening is mentioned in the text, it would be helpful to know whether consent for both TB and HIV screenings was undertaken simultaneously or separately and were there instances where TB screening was acceptable but not HIV or vice versa

5. In a process driven paper such as this, it would be desirable to know the process in qualitative terms viz. where were the screenings undertaken, how long did it take to do the screenings and challenges during screening.

6. Line 160-163 is confusing. Do the authors imply that follow up of the confirmed TB patient was necessitated for a retrospective classification. All the more reason for the work flow to be accurately described.

7. The lack of uniformity in confirming TB diagnosis needs to be explained and the implications in computing prevalence if any

(lines 216-218)

8. An interesting but surprising finding is the minimal amount of TB detected in migrants. Were these recent or long term migrants and what was the disease status in the areas they migrated from.

9. Again a similiar scenario in homeless persons generally considered to be one of the most vulnerable. Where did these persons live: on the streets or were they institutionalized/ congregated in any way. Descriptions are necessary to understand the observations.

10. The observations do not have break up of findings as per age and gender though the information is partially provided in the tables with minimal stratification. Difficult to vizualize whether the high prevalence figures computed define certain subpopulations in the cohort. The findings need to be analyzed and represented in much more depth.

11. The discussion provides valuable insights and touches upon the risk to to HCW, the need to converge services for TB HIV (and NCDs also as in other parts of the world) and the overall lack of infection control. The low prevalence of TB in IDP is well explained. Therefore it would benefit the manuscript in the Results section to include the temporal data of the movement of the displaced persons.

12. The last sentence needs to be better formatted.

13. The message of comprehensive care through service convergence is one that is valuable in many settings and should not be ignored.

Reviewer #2: Comments to the authors:

The cross-sectional study by Dememew ZG et al. deals with the prevalence of TB and HIV in few hot-spots of Ethiopia. Ethiopia is among the top 10 high TB/HIV coinfection prevalence state. Studies are needed which identify TB or TB/HIV coinfection hot spots and also suggest measures to curb these deadly human menaces. Within the identified hot spots, they selected vulnerable population educated them about the importance of TB screening and HIV testing prior to the recruitment of the study. Although, the incidence of TB has dropped in most regions of the world including Ethiopia. However, TB with HIV coinfection is largely prevalent in vulnerable or socially excluded populations and high-risk settings. The studies are required which keep track vulnerable populations and putting them at forefront of interventions to effectively diagnose all the TB cases as well as TB/HIV coinfection. In the current study the authors detected more HIV and TB cases among vulnerable and key populations in the hotspot settings compared to the general population. This study also suggests the importance of community-based collaborations of TB and HIV programmes. The study is well designed, executed and conclusions drawn are well supported by results. The manuscript is very well written and the information provided will greatly help in the management and diagnosis of TB as well as TB/HIV coinfection cases in hot spots of Ethiopia. I have no specific reservations and suggestions for the authors. The manuscript could be considered for publication in PLOS one.

6. PLOS authors have the option to publish the peer review history of their article (what does this mean?). If published, this will include your full peer review and any attached files.

Reviewer #1: No

Reviewer #2: Yes: Mohd Shariq

---

## [Author Response · Author response to Decision Letter 0]

9 May 2020

Dear Editor,

Thank you very much for giving us the chance to address the comments and suggesion from the editor and reviewer.

We have tried our best to deal with the commnets.

Best

The authors

---

## [Editor Report · Decision Letter 1]

12 May 2020

The yield of community-based tuberculosis and HIV among key populations in hotspot settings of Ethiopia: a cross-sectional implementation study

PONE-D-20-01766R1

Dear Dr. Dememew,

We are pleased to inform you that your manuscript has been judged scientifically suitable for publication and will be formally accepted for publication once it complies with all outstanding technical requirements.

With kind regards,

HASNAIN SEYED EHTESHAM

Academic Editor

PLOS ONE

Additional Editor Comments (optional):

The manuscript was sent for major revision and Authors have modified the manuscript keeping in mind the comments of the Reviewers. Reviewer 1 has asked some questions and sought clarifications and elaborations on some points and these have been provided by the Authors. Reviewer 2 has No major comments and others have been taken care of. I recommend this manuscript for publication.
---

## [Editor Report · Acceptance letter]

13 May 2020

PONE-D-20-01766R1 

 The yield of community-based tuberculosis and HIV among key populations in  hotspot settings of Ethiopia: a cross-sectional implementation study  

Dear Dr. Dememew:

I am pleased to inform you that your manuscript has been deemed suitable for publication in PLOS ONE. Congratulations! Your manuscript is now with our production department. 

With kind regards,

on behalf of

Prof Hasnain Seyed Ehtesham 

Academic Editor

PLOS ONE